

# Counting complete? Finalising the plant inventory of a global biodiversity hotspot

Martina Treurnicht[1,2,3], Jonathan F. Colville[4,5], Lucas N. Joppa[6], Onno Huyser[7] and John Manning[8,9]

[1] Department of Conservation Ecology and Entomology, Stellenbosch University, Stellenbosch, Western Cape, South Africa
[2] Institute of Landscape and Plant Ecology, University of Hohenheim, Stuttgart, Germany
[3] South African Environmental Observation Network Fynbos Node, Cape Town, Western Cape, South Africa
[4] Kirstenbosch Research Centre, South African National Biodiversity Institute, Cape Town, Western Cape, South Africa
[5] Statistics in Ecology, Environment and Conservation, Department of Statistical Sciences, University of Cape Town, Cape Town, Western Cape, South Africa
[6] Microsoft Research, Redmond, WA, United States of America
[7] Centre for Biodiversity Conservation, Kirstenbosch Botanical Gardens, Table Mountain Fund (WWF-SA), Cape Town, Western Cape, South Africa
[8] Compton Herbarium, South African National Biodiversity Institute, Cape Town, Western Cape, South Africa
[9] Research Centre for Plant Growth and Development, University of KwaZulu-Natal, Pietermaritzburg, KwaZulu-Natal, South Africa

Corresponding author
Martina Treurnicht,
martinatreurnicht@gmail.com

## ABSTRACT

The Cape Floristic Region—the world's smallest and third richest botanical hotspot—has benefited from sustained levels of taxonomic effort and exploration for almost three centuries, but how close is this to resulting in a near-complete plant species inventory? We analyse a core component of this flora over a 250-year period for trends in taxonomic effort and species discovery linked to ecological and conservation attributes. We show that >40% of the current total of species was described within the first 100 years of exploration, followed by a continued steady rate of description. We propose that <1% of the flora is still to be described. We document a relatively constant cohort of taxonomists, working over 250 years at what we interpret to be their 'taxonomic maximum.' Rates of description of new species were independent of plant growth-form but narrow-range taxa have constituted a significantly greater proportion of species discoveries since 1950. This suggests that the fraction of undiscovered species predominantly comprises localised endemics that are thus of high conservation concern. Our analysis provides important real-world insights for other hotspots in the context of global strategic plans for biodiversity in informing considerations of the likely effort required in attaining set targets of comprehensive plant inventories. In a time of unprecedented biodiversity loss, we argue for a focused research agenda across disciplines to increase the rate of species descriptions in global biodiversity hotspots.

## INTRODUCTION

Global biodiversity hotspots are species-rich areas of high conservation priority, including significant numbers of rare and undiscovered species facing increasing threats of extinction (*Myers et al., 2000*; *Giam & Scheffers, 2012*; *Scheffers et al., 2012*). They provide insight into ecological and evolutionary patterns associated with mega-diverse regions (*Allsopp, Colville & Verboom, 2014*) and the taxonomic and conservation efforts required to document and manage this biodiversity (*Cowling et al., 2003*). Key to this is the urgent (*Scheffers et al., 2012*) but challenging (*May, 2011*) necessity for an adequate bio-inventory (*Pimm et al., 2014*).

Projections suggest that we are decades or more away from achieving acceptable bio-inventories for most hotspots and taxonomic groups (*Mora et al., 2011*). Compounding the uncertainty of how, when, or even if we will achieve this is the lack of adequate data on which to base realistic predictions. Most hotspots are historically under-resourced in terms of taxonomic and scientific effort (*Cowling et al., 2010*; *Grieneisen et al., 2014*), and are far short of achieving near-complete inventories for even conspicuous taxa such as flowering plants (*Sobral & Stehmann, 2009*; *Forzza & Baumgratz, 2012*). Real data on the rates of species descriptions from a particular area is a first requirement for assessing the time-span needed for a near-complete bio-inventory.

Recent analyses suggest that the majority of plant species still to be described are to be found within biodiversity hotspots (*Sobral & Stehmann, 2009*; *Joppa et al., 2011*). Although the rate at which species are documented is determined by taxonomic endeavour, the number of active taxonomists and their individual productivity varies greatly across different taxonomic groups (*Joppa et al., 2011*; *Bacher, 2012*; *Joppa, Roberts & Pimm, 2012*). No comparable analyses exist for an entire flora. Analysing trends in species documentation over time for an entire floristic hotspot will permit real-world predictions of the taxonomic effort required to describe the full species complement of the hotspot. These can serve as guidelines for other species-rich areas in estimating the time frames required for meeting inventory targets such as the Convention on Biological Diversity targets for 2020 (https://www.cbd.int/gspc/objectives.shtml). These estimates can guide the assessment of possible alternative conservation strategies that are less reliant on near-complete species inventories (*Grantham et al., 2009*; *Cowling et al., 2010*; *Forest et al., 2015*).

We focus here on the Cape Floristic Region (CFR, South Africa), one of the smallest (ca. 91,000 km$^2$) of the 25 biodiversity hotspots first identified by *Myers et al. (2000)*, with a flora that is arguably one of the best known among the botanical hotspots. The CFR has been the subject of almost three centuries of intense botanical focus, with a current total of ca. 9 400 recognised species of vascular plants and >68% regional endemism (*Manning & Goldblatt, 2012*). Due to its small size and long legacy of exploration and taxonomic effort, the inventory of vascular plants for the CFR appears to be effectively complete, although new species are still being discovered and described annually (*Manning & Goldblatt, 2012*; *SANBI, 2015*). In this study, we analyse rates of plant species discovery and associated taxonomic effort in the CFR over a 250 year period (1753–2012), utilising techniques similar to those employed by *Joppa, Roberts & Pimm (2011a)* and *Joppa, Roberts & Pimm*

*(2011b)* . We apply these on a subset of >2,400 'Cape clade species' (*sensu Linder, 2003*), representing groups that are centred in the CFR and that also contain a disproportionally high number of endemic and threatened taxa (*Raimondo et al., 2009*). This allowed us to search for trends in species discovery and taxonomic activity in the study region. We also explored trends linked to abundance, ecology, and conservation status. Finally, we considered whether it is possible to estimate how many 'missing species' (*Solow & Smith, 2005*) remain in the CFR and by what date we can expect to identify them.

## MATERIALS AND METHODS

We analysed data for 2,434 species selected from the 33 'Cape clades' identified by *Linder (2003)* as "*…those clades that have had most of their evolutionary history in the Cape Floristic Region*" (CFR; Table S1). These clades represent lineages for which we expect the highest rates of description of new species in recent decades as a natural consequence of the high proportion of local endemics and fire ephemerals in the CFR (*Linder, 2003*; *Webb, Slik & Triono, 2010*; *Manning & Goldblatt, 2012*). We restricted our selection within the clades to genera that (i) have been comprehensively and relatively recently monographed and (ii) included representatives from a wide spectrum of families and growth forms. Only currently accepted species as listed in *Manning & Goldblatt (2012)* were accepted. Our selection comprises 55% of Cape clade species and 26% of all vascular plant species recorded for the CFR.

We determined the date of publication of the protologue for each species, commencing with the publication of *Species Plantarum* (*Linnaeus, 1753*) and terminating with the publication of *Cape Plants* (*Manning & Goldblatt, 2012*). Data on species habit and growth form were culled from *Manning & Goldblatt (2012)* and from PRECIS (National Herbarium **Pre**toria **C**omputerised **I**nformation **S**ystem; *Germishuizen et al., 2006*). Species were classified as annuals, geophytes, graminoids, herbaceous perennials or shrubs. None of the species in the study group were trees. Species distributions were summarised following the phytogeographic centres (*sensu Brown & Stuart, 2009*) as given in *Manning & Goldblatt (2012)*. Species were considered to be 'local' if present in a single phytogeographic centre and 'widespread' (or non-local) if present in more than one phytogeographic centre.

We used sample scripts and functions, readily available online (from *Joppa, Roberts & Pimm, 2011a*), to analyse trends in species discovery and taxonomic activity over ca. 250 years (1753–2012). More specifically, we show "moving average functions" (*sensu Joppa, Roberts & Pimm, 2011a*) described over five-year intervals for (i) the total number of species described, (ii) the cumulative number of species, (iii) the number of taxonomists involved in describing species and (iv) species described per taxonomist. The "number of taxonomists involved in describing species" effectively represents the "taxonomic effort" whereas the measure "species per taxonomist" represents the "catch" or "taxonomic efficiency" (*sensu Joppa, Roberts & Pimm, 2011a*; *Scheffers et al., 2012*). These measures split each unique taxonomic identity (often consisting of multiple taxonomic authors) into individual taxonomic names and accounts for species described by more than one taxonomic author. We then additionally show the (v) relative contribution of each

increment to the flora by dividing each accumulated five-year subtotal of the species described by the cumulative total number of species described over the entire study period (i.e., proportion new species (%) described). Finally, we explore the cumulative number of species by ecological attributes, i.e., (vi) growth form and (vii) phytogeographic centre (a proxy for abundance; from *Manning & Goldblatt, 2012*). All analyses were performed in R version 3.3.1 (*R Development Core Team, 2016*).

## RESULTS

The number of vascular plant species described from the CFR for the 250 year period from 1753–2012 has fluctuated widely over any given five-year increment, from a high of 106 species described in 1830–1835 to a low of 9 species described during 1870–1875 (Fig. 1A; Table S2). The post-World War II period (1945–1990) is characterised by a relatively steady rate in species description of between 48 and 76 species per five-year increment (Fig. 1A; Table S2). Peaks in the latter period represent the dates of publication of significant generic revisions or monographs (e.g., *Goldblatt, 1978*; *Bond & Goldblatt, 1984*; *Linder, 2003*). Overall, however, species in the sample group were described at a remarkably consistent rate over the entire study period (Fig. 1B), with at most only a very slight positive trend in number of species described ($x = 46.51 \pm 20.03$) per increment.

The number of publishing taxonomists per increment ranged from two to 17 with a very slight trend towards more taxonomists over time (Fig. 1C; Table S2), balanced by a corresponding decline in the number of species published per taxonomist (Fig. 1D; Table S2).

Naturally (from Figs. 1A and 1B), the number of new species described each half decade represents a diminishing proportion of the total number of species described over the entire period (Fig. 1E) but the relationship is strongly logarithmic, with a large proportion (41%) of species described within the first century of taxonomic study (1753–1850). Thereafter the additional species contributions stabilised at a modest incremental increase of 3% for the period 1850–1985, with a further decline to the current level of 1.36% (1990–2012; Table S2).

Rates of description of species remained relatively consistent from 1753–2012 regardless of growth form (Fig. 1F), with shrubby species described at a higher rate than other growth forms. In contrast, analysis of the cumulative species by species distribution reveals a critical difference between local and widespread (or non-local) species (Fig. 1G). Local and non-local species accumulated at comparable rates over the period 1753–1930, with a consistently greater number of widespread species being named. The period after 1950, however, documents a deceleration in the description of non-local taxa and a marked acceleration in the description of local taxa. From 1950 onwards, the number of local taxa described per half decade exceeds the number of widespread taxa described in the same period, with a widening difference between the two trends (Fig. 1G).

## DISCUSSION

Our study shows that there has been a sustained level of taxonomic interest in the CFR over more than 250 years of active botanical study. Species have been described at a constant

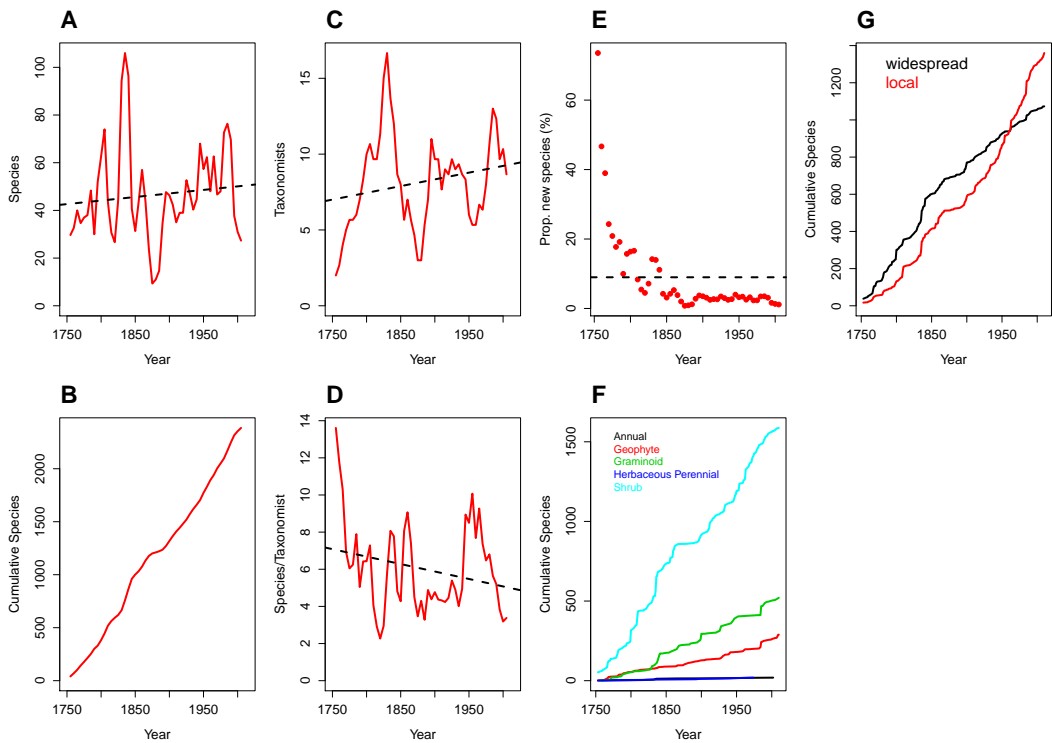

**Figure 1  Trends over time (1753–2012) in species discovery rates and taxonomic effort in the Cape Floristic Region, South Africa.** (A) Total number of species described, (B) cumulative number of species, (C) number of taxonomists involved in describing species ("taxonomic effort") and (D) species described per taxonomist ("taxonomic efficiency"). Plotted lines are moving average functions (*sensu Joppa, Roberts & Pimm, 2011a*) calculated at five-year time intervals across the study period (i.e., 1753–2012). (E) The proportion (%) of new species described per five-year interval, (F) cumulative number of species per growth form (annual [19], geophyte [289], graminoid [520], herbaceous perennial [19] and shrub [1587]) and (G) cumulative number of species by phytogeographic centre (widespread and local; see main text for details) across the study period. Trend lines (dashed black) in (A, C, D) are based on linear model fits whereas the trend line in (E) is based on a mean rate of 8.98 species described.

rate over this entire period, reflecting a sustained level of taxonomic output by a relatively stable number of active taxonomists (Figs. 1C and 1D). From this, we conclude that there is a finite number of species that can be processed by any taxonomist over a fixed period in their working life. Specifically, we suggest that there is a maximum limit to the 'productivity' of any taxonomist, a measure that we term here the 'taxonomic maximum', determined by several factors (e.g., taxonomic group, personality, institutional support, etc.). On the assumption that botanical study in the CFR has proceeded at or near the taxonomic maximum (Fig. 1), we conclude that it is unreasonable to expect an increase in individual taxonomic output under current technologies. The operational concepts applied by practising taxonomists, although seldom if ever explicitly outlined, may also influence a taxonomist's output. We can accept, however, that the species concepts applied by taxonomists in this study are essentially morphological, sometimes in concert with ecological considerations, and there seems little doubt that this is overwhelmingly the case elsewhere in megadiverse regions.

The CFR has been the subject of botanical exploration since the fifteenth century, with a period of particular intensive discovery and documentation in the late eighteenth century (*Glen & Germishuizen, 2010*). This early knowledge was consolidated and expanded in the nineteenth century in the publication of the *Flora Capensis* (*Harvey, Sonder & Thiselton-Dyer, 1859–1933*). Taxonomic activity over this period, based on typological principles and practised largely by non-resident scientists from foreign institutes in Britain and Europe, resulted primarily in a proliferation of names. The early part of the twentieth century was a period of more intensive study associated with the establishment of the region's two primary taxonomic institutes, the University of Cape Town and Kirstenbosch Botanical Garden, with a resident staff. Botanists at these institutes studied the local flora in the herbarium and in the field, and also encouraged collecting among local amateurs. Current civil society programmes continue this valuable contribution in the CFR (*SANBI, 2015*). Similar investments have also made valuable contributions to species discoveries in other hotspots (e.g., the Southwest Australian Floristic Region (SWAFR; *Hopper & Gioia, 2004*)). It should not be overlooked that the actual taxonomic effort expended in the early twentieth century is obscured in our analyses by the fact that we have not included the number of names that disappeared into synonymy during each half decade. This is a significant part of the effort necessary for an accurate bio-inventory.

Professional taxonomists remain the critical resource and our findings suggest that the most effective way of increasing 'taxonomic effort', and thus the rate at which any flora is catalogued, is by increasing the number of active taxonomists (e.g., *Godfray, 2002*; *Bacher, 2012*). Our analysis of the activity of taxonomists also reflects a trend towards multi-authored species, as discerned by *Joppa, Roberts & Pimm (2011a)*.

The proportionally higher number of shrubby species in our study group reflects the predominance of this habit (54% of the flora) in the CFR study region (*Goldblatt & Manning, 2002*; *Linder, 2003*). Other categories are also broadly consistent with their representation in the flora as a whole but reflect sampling bias in the study group (notably graminoids, which are overrepresented in the sample).

The documentation of the CFR flora has proceeded as a logarithmic function, with a long tail representing an incremental addition to the floristic inventory of 1–5% of the sample group every five years. It is an astonishing finding that the description of species from the CFR has continued at essentially the same rate since the documentation of the flora started over 250 years ago. In essence, therefore, the number of newly described species appears to continue to increase at a rate of 20–60 species every five years. Effectively, however, the 'missing species' thus comprise an insignificant proportion of those already described. Based on the fact that some half of the CFR species are members of the Cape clades used to generate these trends, we estimate that the 'missing species' remaining in the CFR constitute <1% of the total flora, which falls far below the predicted numbers of missing plant species (ca. 15%) for other hotspots (*Joppa, Roberts & Pimm, 2011b*; see also *Webb, Slik & Triono, 2010*; *Laurance & Edwards, 2011*; *Scheffers et al., 2012*). For most practical purposes, therefore, the botanical diversity in the CFR can be considered to be adequately

known. Critically, however, the 'missing species' in the CFR are likely to be range-restricted, local endemics that are thus especially vulnerable to extinction (e.g., Fig. 1G).

The Cape flora is characterised by high numbers of local endemics, reflected in the high levels of beta and gamma diversity across the region (*Cowling, Holmes & Rebelo, 1992*; *Goldblatt & Manning, 2000*). As we might expect, the distinction between local and more widespread species appears to be the primary determinant of the likelihood of discovering new species in the CFR since 1950. This conclusion is universal to global biodiversity hotspots (*Scheffers et al., 2012*).

The number of local endemics in the CFR described in the past 50 years is high enough to offset the declining rate at which more widespread taxa are being discovered. This finding has significant implications for conservation in the CFR, and likely also other global hotspots, by confirming that locally endemic taxa are among the last to be discovered. This increases the risk that they will be driven to extinction before being documented since the transformation of species-rich natural habitats continues at alarming rates in both Mediterranean- and tropical hotspots (e.g., *Giam & Scheffers, 2012*; *Pimm et al., 2014*). To mitigate this it is necessary to maintain the level of taxonomic effort and to increase the level of exploration of these hotspots.

The species accumulation trends that we have documented here demonstrate that the CFR is in an enviable position among global biodiversity hotspots in that its botanical diversity is now effectively documented. We are only able to reach this conclusion because the region has been extensively and intensively studied over a period of almost three centuries with a relatively constant taxonomic effort. The situation in other hotspots and among other taxonomic groups (e.g., *Picker, Colville & Van Noort, 2002*) is seldom so favourable. The development of comprehensive conservation assessments of individual species (e.g., *Raimondo et al., 2009*) as a guide to decision-making on how best to invest scarce conservation resources is only possible once near-complete species inventories exist. Such inventories depend on a combination of exploration and documentation. The first objective of the GSPC is to ensure that "*plant diversity is well understood, documented and recognised*" (GSPC, https://www.cbd.int/gspc/objectives.shtml) and our CFR case study directly illustrates that obtaining such a basis for biodiversity estimates takes both time and taxonomic investment. Similar investigations are needed in other Mediterranean hotspots (e.g., SWAFR, California) that have experienced extensive botanical exploration to allow comparative estimates on plant inventories among these hotspots and thus contribute to the foundational objective of the GSPC.

Even if most plant species in the CFR have been named, we largely lack information on abundance, distribution, ecology and other attributes of species that affect their conservation (e.g., *Raimondo et al., 2009*; *Costello, Vanhoorne & Appeltans, 2015*). Recent national threatened species programmes, local citizen science projects, and modern taxonomic revisions have contributed significantly to both species discoveries and unknown locality records of species in the CFR (e.g., *Raimondo et al., 2009*; *Manning & Goldblatt, 2012*; *SANBI, 2015*). Maintaining or even increasing the funding that supports such collaborative efforts is urgent, especially in regions that may not have a long legacy of botanical exploration. Key to filling these gaps in global biodiversity hotspots is

thus to increase collaboration amongst international taxonomists, maintain current taxonomic effort, and increase expertise by including non-specialists (*Costello, Vanhoorne & Appeltans, 2015*), so as to shrink 'the pool of missing species' (*sensu Joppa, Roberts & Pimm, 2011a*). Disproportionately many undiscovered species in hotspots may remain as cryptic endemics with complex morphological differentiation, requiring a combination of specialist taxonomic input, trained technicians and novel techniques (e.g., *Ficetola et al., 2008*) across disciplines (taxonomy, systematics, molecular phylogenetics, population genetics, and ecology; *Webb, Slik & Triono, 2010*; *Scheffers et al., 2012*). The CFR case study we have presented here provides a valuable but rare real-world dataset that other biodiversity hotspots can use to estimate the resources in time, effort, taxonomic output and potential alternative strategies that would be needed to achieve adequate documentation of plant species.

## ACKNOWLEDGEMENTS

We are grateful to Domitilla Raimondo, Lize Von Staden (Threatened Species Programme, South African National Biodiversity Institute (SANBI)) and Ismail Ebrahim (SANBI) for insightful discussions; Ilva Rogers and Les Powrie (SANBI) for assistance with data.

### Funding

Martina Treurnicht received funding from the Table Mountain Fund (TMF, WWF-SA), the Department of Science and Technology - National Research Foundation (DST–NRF, South Africa) internship programme, and is currently supported by Stellenbosch University and the South African Environmental Observation Network (SAEON, Fynbos Node). Jonathan F. Colville is supported by a National Research Foundation of South Africa RCA-Fellowship (Grant 91442). The funders had no role in study design, data collection and analysis, decision to publish, or preparation of the manuscript.

### Grant Disclosures

The following grant information was disclosed by the authors:
Department of Science and Technology - National Research Foundation (DST–NRF, South Africa).
Stellenbosch University.
South African Environmental Observation Network.
National Research Foundation of South Africa RCA-Fellowship: 91442.

### Competing Interests

The authors declare there are no competing interests.

### Author Contributions

- Martina Treurnicht conceived and designed the experiments, analyzed the data, contributed reagents/materials/analysis tools, wrote the paper, prepared figures and/or tables, reviewed drafts of the paper.

- Jonathan F. Colville analyzed the data, contributed reagents/materials/analysis tools, wrote the paper, prepared figures and/or tables, reviewed drafts of the paper.
- Lucas N. Joppa analyzed the data, contributed reagents/materials/analysis tools, prepared figures and/or tables, reviewed drafts of the paper.
- Onno Huyser conceived and designed the experiments, reviewed drafts of the paper.
- John Manning conceived and designed the experiments, contributed reagents/materials/analysis tools, wrote the paper, prepared figures and/or tables, reviewed drafts of the paper.

### Data Availability

The raw data has been supplied as a Supplementary File.

### Supplemental Information

Supplemental information for this article can be found online at http://dx.doi.org/10.7717/peerj.2984#supplemental-information.

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
