# Peer review of "Counting complete? Finalising the plant inventory of a global biodiversity hotspot"

_PeerJ, doi:10.7717/peerj.2984_

## Round 0.1 · original submission · Minor Revisions

All three reviewers were enthusiastic about your paper and recommended publication. However, Reviewers 1 and 3 have made some useful suggestions for improving the manuscript. Expanding your Discussion to include likely patterns in other Mediterranean hotspots will usefully increase the scope of your paper, and I recommend that do this.

·

Basic reporting

Counting Complete? by Treurnicht et al. is a well written analysis of the long history of plant inventory of the Cape Floristic Region (CFR). The English is clear and concise, the writing is succinct and the background review thorough. I had no difficulty in following the arguments put forth by the author team. One only figure was offered, and it too was a simple suite of graphs that illustrated the major findings of the article. In short, this was a very well thought out and well written piece.

Experimental design

As is pointed out in the article, the CFR is exceptionally well known for a mega diverse flora. Thus the experimental design was not hampered in any way by a lack of information, nor by any conflicting information about species richness or taxonomic uncertainties. I very much liked that the overall message and intent of the analysis was to further conservation efforts both in South Africa as well as in other hotspots. I also appreciate that it was appropriately tied to the Global Strategy for Plant Conservation. There is nothing to criticism in this topic - the analysis was well done.

Validity of the findings

The findings by Treurnicht et al were, as required, robust and statistically sound. And although the work did not require a particularly sophisticated statistics, sometimes this kind of simply inquiry is particularly meaningful. There were a couple of additional topics I would have hoped the authors addressed, however, rather than staying strictly to commenting on the results of the analysis for the Cape Flora. I think that small speculation into what these results signify for other extra-tropical hotspots such as the Southwest Australia Floristic Region (SWAFR) or the Mediterranean Basin flora would have provided a deeper context for the findings. Another issue that might have been worth addressing, and thus given greater weight to the findings relative to the rate of species descriptions is the prevailing species concepts for the CFR. Obviously species concepts influence taxonomic distinctness and recognition, and providing clarity or perhaps methodological boundaries for the work of taxonomists working in the CFR could be important to establish the framework of plant descriptions across the decades (centuries!). Lastly, I'm alittle uncomfortable with the idea of 'taxonomic maximum' because my own experience suggests that productivity of any taxonomist is a function of many things (e.g., taxonomic group, the scientist's personality, institutional support, where published?, appropriate species concept(s), etc), many of which work in concert (and are out of the ability of the scientist to control) to influence 'productivity.' I think this idea might need a bit more fleshing out.

Additional comments

I have a few other comments to make for the authors. Your paper is solid and straight forward, but it wasn't necessarily one that makes me think hard about where might you go next. I'll address that in a minute. A couple of minor things to fix.
First, I couldn't find Glen & Germishuienzen 2010 cited in the paper. Maybe I missed it, or maybe it wasn't cited. Please check for it again.
Second, you cite Harvey, 1869-1900 and it should be cited Harvey, Sonder & Thistle-Dyer, 1869-1900.
Third, Godfray 2002 is out of alphabetical order in your references section, and last for the little things,
Fourth, in Figure 1(f), I didn't see the line for herbaceous perennials. Unless the lowest line is actually purple, and then I can't see the annual line. Or are they overlapping?
OK, on to adding some sizzle to the findings, in addition to my comment about adding a statement concerning guiding species concepts for the taxonomic work in the CFR. Obviously the CFR is one of five recognized Mediterranean-type climate regions of the world. I'm not a real advocate of lumping them all together as a single, global phenomenon (instead, finding their differences real and meaningful), but they are often discussed together. Perhaps some reasoned speculation about what the findings mean for other MTEs might spur on this kind of analysis in other places - e.g., California, Western Australia. Prof Steve Hopper did the cumulative species description analysis some time ago for the SWFR, and perhaps could have been consulted or maybe even SWFR included in the paper? The authors do suggest that these insights are useful for other hotspots (e.g., locally endemic species are usually the last to be described) but a little deeper commentary on this issue would have enhanced the paper. In the end, comparative work, which is possible because some of these MTEs are well known, like California and the Mediterranean Basin, is often more interesting and potentially insightful than an analysis of a single region or issue. Nonetheless, this is a well done analysis and can stand alone.

·

Basic reporting

This is an entirely straightforward analysis on which I have no serious comments.

The authors correctly point out that the Cape Flora is both a biodiversity hotspot and a place that is exceptionally well-known. It's thus a perfect case study to look into the details about the rate of plant species descriptions. Such details are most informative in our understanding of how many species there might be in the world and where they might be found. ("About 15% more" and "in the hotspots" seem to be the answers, but close examinations of well-known, endemic rich floras are key to assessments.)

As this paper shows, there is a steady stream of new species, particularly those species with narrow geographical ranges. This more than hints at what is like likely to be the pattern elsewhere.

In short, this is simple, crisp and add useful insights into an important question.

Experimental design

No further comments to add.

Validity of the findings

No further comments to add.

Additional comments

No further comments to add.

Reviewer 3 ·

Basic reporting

This manuscript investigate trends in the discovery and description of new species of plants in the Cape Floristic Region. This region is probably one of the best in the world to evaluate these trends due to the extensive existing knowledge of its flora. It is well-structure, contains sufficient background information for readers not familiar with this system to comprehend it, and figure provided is relevant and clear. There are no problems with the English.

Experimental design

As far as I am aware, this type of analysis has not been performed for the CFR to date. The methods the authors used are generally well-described and appropriate for the questions they address. I have only two minor comments/questions:
- I understand that "the number of taxonomists involved in describing species" is in total, not the number of taxonomists involved in describing a particular species? It is not clear, but maybe I missed this, how the authors accounted for species described by more than one taxonomist.
- As rightly pointed out, the analyses presented here do not account for taxa that "disappeared into synonymy" (L180-182). I wonder if this couldn't be investigated using groups for which this information is available, such as Restionaceae and Poaceae using resources such as IPNI and the World Checklist of Selected Families which have synonyms as well as accepted names?

Validity of the findings

The study is well-executed and present valuable results. It is in fact somewhat surprising that this kind of analyses have not been performed before on this flora.

Additional comments

Minor comments:
L85: Delete "highly".
L93: Why do you expect that the clade which have mostly diversified in the CFR will have the highest rates of description of new species? Wouldn't more depend on if they have been worked on more than their diversification rates?
L117: "per taxonomists" rather than "by taxonomists"?
L148: Since most of the clade you used are form mostly of shrubs, wouldn't you expect these to be described at higher rates?
L154-156:I guess this would be expected, no? Widespread taxa would be found first and thus described before local endemics.
L228-229:It is great that the CFR has a near-completed species checklist, but I don't agree that only when this is achieved should we developed conservation assessments.
L236: "for example" instead of "(e.g.)".
L356: Maybe better to have "widespread" and "local" on the figure rather than 0 and 1, as in Fig. 1f?

---

## Round 0.2 · accepted · Accept

I am satisfied that you and your team have addressed all of the comments made by the reviewers.